

# Xenobiotic resistance in mosquito eggs: current understanding and data gaps

Uday Turaga[1,2], Steven T. Peper[3], Carlos J. Garcia[1] and
Steven M. Presley[1]

[1] The Institute of Environmental and Human Health, Texas Tech University, Lubbock, Texas,
United States
[2] Department of Biotechnology, Koneru Lakshmaiah Education Foundation, Vaddeswaram,
Andhra Pradesh, India
[3] Anastasia Mosquito Control District, St Augustine, Florida, United States

## ABSTRACT

Of all mosquito life stages, the egg continues to be the least understood and most vulnerable relative to exposure to external environmental stressors. The propensity of mosquitoes to lay eggs in or near aquatic environments exposes them to a variety of xenobiotic compounds. Owing to their increased use, two xenobiotics, antibiotics and insecticides are increasingly being detected in aquatic ecosystems. Both antibiotics and insecticides are known for their ovicidal effects. Prior to the formation of the serosal cuticle, mosquito eggs are potentially exposed to antibiotics and insecticides due to the permeability of the egg membrane. This short review attempts to summarize the current understanding and to identify the data gaps pertaining to the exposure of mosquito eggs to xenobiotics. The role of male mosquitoes in the propagation of xenobiotic resistance, something that's been sparsely studied, is also discussed. Additionally, we address the implications of these data gaps relative to the overall objectives of vector control and public health.

## INTRODUCTION

Mosquitoes are holometabolous insects, undergoing a complete metamorphosis, developing from eggs to larvae, to pupae, and finally to adults. The mosquito egg is a closed system within which all nutrients required for embryogenesis are present, except for oxygen and water. Eggs depend on their immediate environment for oxygen and water (*Farnesi et al., 2015*; *Prasad et al., 2023*). Freshly laid mosquito eggs are highly susceptible to water loss, a significant threat to their viability (*Panfilio, 2008*). The permeability of eggs during the beginning stages of embryogenesis accommodates this requirement (*Mundim-Pombo et al., 2021*). However, the permeability of eggs to water decreases during the course of development/embryogenesis (*Vargas et al., 2014*). The increased use of xenobiotics like antibiotics and insecticides has resulted in their increased presence in aquatic ecosystems, and the dependence of mosquito eggs on water for development increases their likelihood of exposure to these xenobiotics.

The present short review attempts to summarize the current understanding pertaining to the physiology of eggs and how it makes them prone to exposure to antibiotics and

Corresponding author
Steven M. Presley,
steve.presley@ttu.edu

insecticides. The formation of serosal cuticle (SC) and how it protects the egg from exposure to xenobiotic stressors is discussed. The effects of antibiotics and insecticides on the viability of eggs and adult mosquitoes, is examined. Finally, scientifically meaningful data gaps are identified with regards to how the eggs and adult mosquitoes are surviving exposure to antibiotics and insecticides. This includes the likelihood of vertical transfer of antibiotic and insecticide resistance genes to eggs, and microbiome-mediated resistance to antibiotics and insecticides in adult mosquitoes. The often neglected and understudied role of male mosquitoes in the propagation of xenobiotic resistance is also addressed, and the implications of these knowledge gaps relative to the overall success of vector control and public health are assessed.

Mosquitoes are ubiquitous at the interface of many different ecosystems, and their role in vectoring infectious diseases in humans and animals is well documented. In this regard, it is important to understand how mosquitoes are surviving exposure to xenobiotic stressors. This applies not just to adult mosquitoes, but mosquitoes in all life stages. The egg remains the most vulnerable and least understood life stage in a mosquito. The knowledge gaps identified in this review are of significant relevance to the scientific communities trying to understand the phenomenon of xenobiotic resistance in mosquitoes in all life stages, especially eggs. These include regional vector control programs, and insecticide resistance research groups trying to decipher the presence, persistence, and propagation of insecticide resistance in various ecosystems. Understanding the transovarial and transstadial transmission of antimicrobial resistance is also important to decipher the role of mosquitoes in emerging infectious diseases. Finally, mosquito eggs and larvae are a constant presence at the bottom of the aquatic food chain where many small fish consume them. The knowledge gaps identified in this review are also of interest to research groups that study the presence and propagation of antimicrobial resistance along the food chain.

## SURVEY METHODOLOGY

A thorough and comprehensive review of existing literature was conducted using commonly used search engines like the Google Scholar, Web of Science, *etc*. Only published and peer reviewed literature was considered. The most referred search terms included, but were not limited to, "insecticide resistance in mosquito eggs", "antibiotic resistance in mosquito eggs", "vertical/transovarial transmission of xenobiotic resistance in mosquitoes", "mosquito microbiome", "microbiome mediated xenobiotic resistance in mosquitoes", "mosquito eggs", *etc*. Preference was given to recent literature (after 2000's) but older literature was also referred to, as needed.

Studies to a very limited degree have investigated the phenomenon of xenobiotic resistance in insect eggs. However, any literature pertaining to oviposition and mechanisms of xenobiotic resistance in other insect species, especially in eggs, cannot be attributed to mosquito eggs for the reasons discussed in the following section.

## Egg physiology and exposure to xenobiotics (antibiotics and insecticides)

Egg laying is an important aspect of female insects' reproductive biology. Considering its significance, the "when" and "where" of egg laying is critically regulated and conserved in insect species. Insect species select the location to oviposit depending upon its ability to support the process of larval development. A non-exhaustive and generally accepted list of oviposition sites in insects includes soil (beetles, flies, *etc.*), plants (phytophagous insects), vertebrate wounds (flies), insect larvae (parasitic wasps, leafhoppers), body of insects or spiders (parasitic wasps), decaying organic matter (saprophagous insects), and water (mosquitoes) (*Cury, Prud'homme & Gompel, 2019*). The behavioral tendency of insect species, like mosquitoes, to return to similar egg laying sites over and over across generations is termed oviposition constancy. Oviposition constancy in insects is often a combination of innate and learnt preferences and helps females minimize their efforts being spent on unsuccessful oviposition (*Nataraj, Hansson & Knaden, 2024*). The inclination of gravid mosquitoes to select water bodies for oviposition, unlike many other insect species, is an important factor that increases the likelihood of exposure of eggs to xenobiotics. This makes the understanding of the survival mechanisms of mosquito eggs before the formation of SC all the more relevant.

As is the case with all insects, the production of eggs in mosquitoes is a complex and carefully regulated process. The initiation of follicle development by an accumulation of vitellogenin yolk proteins begins after a female mosquito takes a blood meal from a host. Most of the structural components of an eggshell are secreted by a single layer of follicular epithelial cell surrounding the oocyte. At the proteomic level, eight different proteins involved in the fecundity, melanization, and viability of mosquito eggs have been identified. These proteins include Nasrat, Closca, Polehole, Nudel, DCE2, DEC4, DCE5, and CATL3. Nasrat, Closca, Polehole, Nudel, and DCE2 proteins play a critical role in maintaining the integrity of the eggshell. Other proteins include structural proteins, enzymes, odorant binding proteins, and uncharacterized proteins of unknown functions (*Isoe et al., 2019, 2023*).

At the microscopic level, *Mundim-Pombo et al. (2021)* investigated the timeline of embryogenesis in detail. At the proteomic level, a non-exhaustive list of eggshell proteins and their functions is delineated in Table 1 (*Marinotti et al., 2014*).

A recent comprehensive bioinformatic analyses of putative protein-coding sequences of *Aedes*, *Culex*, and *Anopheles* genomes has resulted in the identification of another protein, *i.e.*, the Eggshell Organizing Factor 1 (EOF1). The authors hypothesized that the EOF1 protein has conservatively evolved within the Culicidae family and affects eggshell formation and melanization (*Isoe et al., 2019, 2023*).

At the phenotypic level, the formation of the eggshell and SC during mid-to-late stages of embryogenesis plays a crucial role in the viability of the developing embryo. The formation of SC imparts desiccation resistance and ensures the survival of mosquito eggs during dry conditions. Considering the potential changes in climate globally including prolonged drought periods, the significance of SC in mosquito eggs cannot be

**Table 1 Eggshell proteins and their functions.**

| Eggshell protein | Examples & functions |
|---|---|
| Vitelline membrane proteins (VMPs) | Vitelline membrane is the inner layer of the eggshell. VMPs 15a-1, 15a-2 and 15a-3 potentially serve as a bridge between the eggshell layers |
| Chitin-binding proteins (CBPs) | CBPs provide rigidity to the mosquito eggshell |
| Odorant binding proteins (OBPs) | OBPs serve as the structural components of the chorion intermediate layer |
| Cysteine-rich proteins (CRPs) | CRPs function as the structural components of the external chorionic layer of the eggshell |
| Enzymes | Phenol oxidases-melanization of the chorion; Laccases-eggshell tanning and sclerotization; Peroxidase-eggshell formation; Transglutaminases-cuticle morphogenesis and sclerotization; Dopachrome-conversion enzyme-eggshell formation |
| Cellular and structural proteins | Actin, myosin, heat shock proteins, and serine proteinases |

underestimated. Even at the species level, the cuticle has been highly preserved among insects over the course of evolution. The cuticle offers protection by preventing the penetration of external compounds, like xenobiotics. The outermost layer of a cuticle is the epicuticle and is composed of hydrocarbons, proteins, and lipids. The lipids offer a hydrophobic barrier preventing the entry of hydrophilic compounds like insecticides (*Balabanidou, Grigoraki & Vontas, 2018*).

During early embryogenesis, the eggshell is only composed of exochorion and endochorion. The exochorion and endochorion are maternally derived and are secreted by the follicle cells. At this stage, the eggshell is highly permeable to water, and exposure to xenobiotics through water. The propensity of mosquitoes to oviposit in or near water sources exacerbates this exposure. The zygotically derived extraembryonic serosal cells secrete the SC over the course of embryogenesis. The SC eventually becomes the third and innermost layer and decreases the water permeability of eggs (*Farnesi et al., 2017*; *Muthukrishnan et al., 2022*; *Isoe et al., 2023*).

An important component of eggshells is a homopolymer of N-acetyl-D-glucosamine linked by β-1, 4-glycosidic bonds (*i.e.*, chitin). The biosynthesis of chitin in insects starts with trehalose sugar found in the insect hemolymph and ends with the formation of chitin polymer. Chitin synthases (CHS) are membrane bound enzymes and use the activated "uridine diphosphate-N-acetylglucosamine" to form chitin polymer (*Glaser & Brown, 1957*). Chitin synthase A (CHSA) and chitin synthase B (CHSB) are the two most common chitin synthases found in insect species. Chitin synthase A is responsible for the formation of chitin incorporated in the cuticle, and is expressed mainly in the epidermal cells. Chitin synthase B is responsible for the formation of chitin in the peritrophic matrix, and is expressed in the midgut epithelial cells (*Arakane et al., 2005*; *Zimoch et al., 2005*). Chitin in eggshells offers protection from mechanical stress, dehydration, and xenobiotics (*Muthukrishnan et al., 2022*). The biosynthesis, transformation, and modification of chitin is an important requirement in the process of insect molting (*Tetreau & Wang, 2019*).

Molting is crucial in the normal growth and development of insect species (*Zhang et al., 2014*).

The presence of chitin in eggs, eggshells, and ovaries of mosquitoes has been well documented. Additionally, chitinase activity was observed not just in eggs as they near eclosion, but also in newly hatched larvae (*Moreira et al., 2007*). The chitinized SC decreases the permeability of eggs, preventing exposure to xenobiotics through water. Additionally, the chitinized SC also imparts desiccation resistance to eggs (*Farnesi et al., 2015*).

Desiccation significantly impacts the viability of a mosquito egg. Temperature and humidity are found to be the leading causes of desiccation in mosquito eggs (*Juliano et al., 2002*). Depending on the habitat into which the mosquito oviposits, the eggs of some species of mosquitoes are more prone to desiccation than others and exhibit desiccation tolerance accordingly. *Aedes* species oviposit their eggs in moist habitats where they are more prone to desiccation. *Culex* species on the other hand oviposit in water and are less prone to desiccation (*Suman et al., 2013*).

Increase in temperature accelerates the process of desiccation in mosquito eggs. Previous studies have established a relationship between temperature, the length of embryogenesis, and the acquisition of SC-medicated desiccation resistance. The length of embryogenesis and the formation of SC varies between different species of mosquitoes. At 28 °C, the total length of embryogenesis in *Ae. aegypti* eggs was reported to be 61.5 h post-oviposition. The SC was formed during the complete germ band extension phase at 11–13 h post-oviposition (*Rezende et al., 2008*). At 25 °C, embryogenesis was completed in *Ae. aegypti*, *Anopheles aquasalis*, and *Culex quinquefasciatus* eggs at 77.4, 51.3, and 34.3 h after oviposition, respectively. The SC-mediated desiccation resistance was acquired 14–16 h post-oviposition in *Ae. aegypti*, 9–11 h post-oviposition in *An. aquasalis*, and 10–12 h post-oviposition in *Cx. quinquefasciatus* (*Vargas et al., 2014*).

Table 2 summarizes the duration of embryogenesis and the time the SC forms, at two different temperatures in *Ae. aegypti*.

The time required for the formation of SC and the acquisition of SC-mediated desiccation resistance shortened with an increase in temperature. This is important to ensure the viability of eggs. In adult mosquitoes, a microbiome mediated response to heat stress was observed in *Ae. aegypti*. This involved the enrichment of midgut microbiome with heat-tolerant taxa such as *Bacillus spp.* in response to an increase in temperature (*Onyango et al., 2020*).

From the available literature, it is evident that the formation of SC does not happen for at least a few hours post-oviposition. This suggests that the eggs are vulnerable to exposure to insecticides and antibiotics present in the aquatic environments the mosquitoes lay eggs in. The current understanding of the effects of antibiotics and insecticides on mosquitoes and mosquito eggs is summarized in the next section.

**Table 2 Length of embryogenesis and formation of SC timeframes in *Aedes aegypti*.**

| Temperature | Duration of embryogenesis (h) | Time post-oviposition during which formation of SC occurs (h) | Reference |
| --- | --- | --- | --- |
| 25 °C | 77.4 | 14–16 | *Vargas et al. (2014)* |
| 28 °C | 61.5 | 11–13 | *Rezende et al. (2008)* |

## Effect of xenobiotics (antibiotics and insecticides) on mosquitoes and mosquito eggs-current understanding

A 2018 survey by the World Organization for Animal Health estimated that between 69 and 76 kilotons of antimicrobial agents were used in the year 2018 for veterinary applications alone. The data was collected across 109 participating countries. A crucial finding was that antibiotics accounted for more than 54% of the total antimicrobials intended for use in animals (*World Organization for Animal Health, 2022*). The antibiotic runoff from these sources into nearby water bodies is a significant source of exposure of antibiotics to mosquito eggs.

Mosquitoes are at an increased risk of exposure to antibiotics across all life stages owing to the increased presence of antibiotics in the environment. Eggs of mosquitoes may often hatch in water contaminated with antibiotics, and the larvae and pupae also spend considerable time in water contaminated with antibiotics (*Endersby-Harshman, Axford & Hoffmann, 2019*). Adult mosquitoes ingest minute concentrations of antibiotics during a blood meal if the antibiotics are circulating in the host system (*Gendrin et al., 2015*).

Exposure of mosquito eggs to antibiotics is known to negatively affect embryogenesis and hatching success. A possible explanation is the inhibition of amino acid absorption resulting from exposure to antibiotics. Amino acids play a crucial role in egg production and larval development in mosquitoes (*Ha et al., 2021*). Exposure to antibiotics in the larval stage is also known to affect long-term survival and vector competence in mosquitoes. Exposure to antibiotics in the larval and subsequent stages disrupt the microbiome in mosquitoes (*Garrigos et al., 2023*). The effects of antibiotics on mosquito microbiome include alteration in the composition of microbiota, immune mediated responses, and nutrient absorption (*Ferreira et al., 2023*; *Garrigos et al., 2023*). Additionally, exposure to antibiotics is known to negatively affect the production of eggs in female mosquitoes. Antibiotics inhibit the lysis of red blood cells and decrease the digestion of blood proteins. This deprives the mosquito of essential nutrients and eventually results in a decrease in the number of eggs produced by female mosquitoes (*Gaio et al., 2011*).

On the other hand, the global use of pesticides in agriculture in the year 2022 was estimated to be 3.7 million tons of active ingredient. This represented a 4% increase from the year 2021, a 13% increase from the previous decade, and a 200% increase from the 1990s. Insecticides used around the world include chlorinated hydrocarbons, organophosphates, carbamates, pyrethroids, botanical and biological products. Compared to the 1990s, the most recent decade saw a 48% increase in the use of insecticides (*FAO, 2024*). Most of the insecticides are water soluble and their runoff into water bodies is a

significant source of exposure of mosquito eggs to these insecticides. Exposure of adult mosquitoes to insecticides frequently occurs due to regional vector control programs. Additionally, eggs may also hatch in waters contaminated with run-off insecticides from various domestic and regional vector control operations.

The toxic effects of chemicals used in vector control programs on adult mosquitoes and larvae are well documented, and the ovicidal activity of some of those chemicals has also been studied. Exposure of eggs to insect growth regulators like pyriproxyfen, azadirachtin, and diflubenzuron results in abnormal embryonic development. Pyriproxyfen and azadirachtin arrest embryonic development, and diflubenzuron affects chitin development of the embryonic cuticle (*Suman et al., 2013*). Exposure to sublethal concentrations of cypermethrin has resulted in not only fewer eggs laid by female mosquitoes, but also in reduced hatching success (*Sunday, Kayode & Ashamo, 2016*).

The ovicidal effects of xenobiotics clearly affect mosquitoes at a population level. However, it is not clear if eggs are evolving to combat the exposure to xenobiotics prior to the formation of SC. This is a glaring knowledge gap in mosquito biology that needs to be investigated. The next section attempts to present a rational explanation based on existing evidence that could lead to further research in this direction.

## Xenobiotic resistance in eggs and adult mosquitoes: current understanding and knowledge gaps?

### Antibiotics

The presence of antibiotic resistance genes (ARGs) in mosquito eggs has not been studied yet. However, a study reported by *Hyde et al. (2019)* found that antibiotic-resistant bacteria (ARB) were recovered not only from wild-caught mosquitoes, but also from colony-reared mosquitoes. In their study, colony-reared mosquitoes had no previous exposure to antibiotics. Female mosquitoes are known for their role in vectoring different bacteria and viruses between their hosts. A recent study reported by *Mosquera et al. (2023)* established the role of ovipositing female mosquitoes as mechanical vectors of bacteria, as well as their influence on the microbial composition of their breeding sites. Their results suggested that gravid females predominantly vector bacteria of two genuses, *Bacillus* and *Elizabethkingia*. *Elizabethkingia* is found to be a predominant genus in mosquito breeding sites, and they aid in accelerating larval development. Not surprisingly, *Elizabethkingia spp.* are reported to be a major component of mosquito gut microbiome. Additionally, *Elizabethkingia spp.* are known to harbor an array of ARGs, which may explain the presence of ARGs in mosquito microbiome. Considering the observations of *Onyango et al. (2020)*, it needs to be investigated if females vector the heat-tolerant *Bacillus spp.* to ensure the viability of larvae in hot conditions. Mosquitoes start acquiring their microbiome in the larval stage, and the presence of heat-tolerant *Bacillus spp.* very early in this process enhances their chances of survival amidst increasing temperatures.

The preference of gravid females to inoculate their breeding sites with specific genus of bacteria raises an important question on the role of bacteria in facilitating the development of mosquitoes. Likely explanations include the ability of these bacteria to assist in the breakdown of nutrients that supplement the diet of larvae. They also potentially affect the

pH and redox reactions in the gut environment during larval stages and affect the growth and molting processes. Lastly, these bacteria generate conserved metabolites that act as signalling molecules (*Coon, Brown & Strand, 2016*). Nevertheless, mosquito eggs are sterile, and bacteria is only present on the surface of the eggs, and not inside the eggs (*Chao, Wistreich & Moore, 1963*). The presence of bacteria on the surface of eggs could be a result of transovarial transmission or egg smearing (direct transmission). Another possibility is the contamination of the egg environment by adult mosquitoes (indirect transmission) (*Favia et al., 2007*; *Coon et al., 2014*). Regardless, the role of these bacteria in conferring protection from exposure to xenobiotics prior to the formation of SC has not been studied.

*Mancini et al. (2018)* deciphered the microbiota in the reproductive organs, gut and salivary glands of different mosquito species. A significant finding from their study is that the microbiota in the salivary glands of female mosquitoes is predominantly composed of Proteobacteria. Proteobacteria are known to be one of the most abundant phyla in the human gut microbiota (*Rizzatti et al., 2017*). In tandem, the presence of antibiotic-resistant bacteria in female mosquitoes, and the ability of gravid females to vector bacteria is a potential One Health concern. It leads to an important question whether mosquitoes are vectoring ARB and ARGs between their hosts, in addition to other pathogens. The role of various insects in the acquisition and transmission of antibiotic resistance has been extensively reviewed by *Rawat et al. (2023)*. However, the potential for mosquitoes to acquire and transmit antibiotic resistance has been sparsely discussed, despite their presence at the interface of various ecosystems.

Mosquito-borne bacterial diseases in humans have not been well understood compared to mosquito-borne viral infections. Nevertheless, studies have established the role of mosquitoes as vectors in the transmission of bacterial infections. The direct transmission of *Rickettsia felis* and *Francisella* to healthy mice through the bite of an infected mosquito has been demonstrated *in vitro* (*Dieme et al., 2015*; *Abdellahoum, Maurin & Bitam, 2020*). A crucial finding is the presence of bacterium in the salivary glands for it to be transmitted. This is the very same requirement in the transmission of arthropod-borne viruses (*Laroche, Raoult & Parola, 2018*). Considering the ability of mosquitoes to transmit bacteria once in their salivary glands, it is very likely that mosquitoes potentially transmit antibiotic-resistant bacteria between their hosts. However, ARGs from proteobacteria would need to get from skin to gut for this to be reciprocal transfer.

Additionally, the presence of antibiotic resistance genes in a mosquito raises other important questions: (1) is it attributable to the presence of the microbiota that are harboring ARGs? (2) Is it attributable to the presence of ARGs in the mosquito genome itself? (3) Can the presence of ARGs in the mosquito genome be attributed to the presence of these genes in the eggs of mosquitoes (transovarial/vertical transmission)?

### Insecticides

Considering the ovicidal effects of insecticides, it is important to investigate how mosquito eggs survive exposure during such a vulnerable life stage. Known mechanisms of insecticide resistance in adult mosquitoes include: (1) metabolic detoxification by enzymes;

(2) target site insensitivity; (3) cuticular resistance/reduction of penetration; (4) behavioral resistance (*Liu, 2015*; *Karunaratne et al., 2018*; *Bharathi & Saha, 2021*). Additionally, the role of mosquito microbiota in conferring insecticide resistance to adult mosquitoes is increasingly becoming evident. *Almeida et al. (2017)* reported successfully isolating insecticide-degrading bacteria from the gut of mosquitoes. The isolated bacteria included *Pseudomonas stutzeri*, *Pseudomonas psychrotolerans*, *Arthrobacter nicotinovorans*, *Leclercia adecarboxylata*, and *Microbacterium arborescens*. They found that even when accounting for the natural degradation of target insecticides, the bacteria still degraded 25–75% of insecticides. The presence of bacteria that can degrade insecticides and use the components as a source of carbon has also been identified in the gut of other insects such as grain beetles (*Wang, Wang & Lu, 2022*). However, the microbiota in a mosquito is not acquired until the larval stage. The larvae of mosquito eclose with no microbes in their gut, thus the occurrence and composition of their gut microbiota is highly dependent on the aquatic habitats in which they develop (*Strand, 2018*; *Ranasinghe & Amarasinghe, 2020*). On the other hand, genetic modifications that confer resistance though mechanisms such as target site insensitivity are of little use in the egg stage without a suitable target (*e.g.*, the nervous system). Cuticular resistance/reduced penetration in case of eggs may not come into effect until the formation of the SC. It is possible that enzymes such as cytochrome P450s, esterases and glutathione-S-transferases may play a role in protecting the egg when exposed to insecticides. Another possibility is that the upregulation of genes like CHSA may accelerate the formation of SC to increase the water impermeability in eggs (*Rezende et al., 2008*). Chitin synthase A is not only important in the egg stage, but all life stages of a mosquito. The knockdown of CHSA using small interfering RNA in the larval and pupal stages prevented larval molting, pupation, and adult eclosion. Knockdown of CHSA in the adult stage affected the laminar organization of the mesoderm and formation of pseudo-orthogonal patterns of the large fibers of the endoderm (*Yang et al., 2021*).

If microbiome-mediated insecticide resistance in mosquitoes is indeed another mechanism of resistance, it is important to consider how the absence of microbiota in lab-reared eggs and larvae affects insecticide susceptibility/resistance studies in the laboratory. The microbiome is not acquired until the larval stage. Eggs collected in field ovitraps and hatched in a laboratory setting are devoid of a conventionally field acquired microbiome. Progenies (F1 generation) of lab reared mosquitoes run into the very same issue.

It can be reasonably concluded that there remains a huge data gap pertaining to the survival of eggs to xenobiotics. A likely explanation could be the transovarial transmission of antibiotics and insecticide resistance genes to eggs ensuring their viability. One way to investigate this phenomenon is the extraction of nucleic acids from mosquito eggs. This is followed by screening for the presence of known antibiotic and insecticide resistance genes.

## Role of male mosquitoes in the propagation of xenobiotic resistance

Male mosquitoes are often not considered for any xenobiotic susceptibility/resistance studies. A big reason is their lack of vectorial capacity, and their lack of ability to transmit diseases (*Alphey et al., 2010*). However, a few observations from recent studies and critical

data gaps pertaining to the role of male mosquitoes in the propagation of xenobiotic resistance are highlighted below:

1. *Mains, Brelsfoard & Dobson (2015)* have investigated the use of male mosquitoes to complement the current vector control approaches to deliver larvicides like pyriproxyfen to mosquito breeding sites. They termed this approach, "Auto-Dissemination Augmented by Males (ADAM)". Male mosquitoes either contaminated the breeding sites or cross-contaminated females delivering lethal doses of larvicides, in both the cage and open field trials. Male mosquitoes also demonstrated tolerance to larvicidal doses of pyriproxyfen. Most importantly, it can be inferred that male mosquitoes act as another source of exposure of xenobiotics to mosquito eggs.

2. A recent study by *Ojianwuna et al. (2024)* in Nigeria revealed that male *Anopheles gambiae* mosquitoes are demonstrating resistance to dichlorodiphenyltrichloroethane (DDT) and lambda-cyhalothrin, and male *Aedes aegypti* mosquitoes are demonstrating resistance to lambda-cyhalothrin.

3. The genetic engineering of male mosquitoes is being considered as another vector control approach lately. Two techniques that are being considered for this purpose are the Sterile Insect Technique (SIT) and the Incompatible Insect Technique (IIT). Both SIT and IIT techniques resulted in reduced mosquito densities and incidence of mosquito-borne diseases like Dengue, as reviewed by *Rahul et al. (2024)*.

From the above observations, it can be inferred that male mosquitoes are demonstrating xenobiotic resistance. An important question in this regard is the mode of acquisition of xenobiotic resistance, especially in male mosquitoes. One explanation is the transovarial transmission of xenobiotic resistance genes, which enable both eggs and adults survive exposure to xenobiotics. The presence of xenobiotic resistance genes in the egg also facilitates its survival until the formation of SC. Another data gap is the role of male mosquitoes in the propagation of xenobiotic resistance. If the male mosquitoes are surviving exposure to xenobiotics due to the presence of resistance genes, it is very likely they are also passing them to the next generations. Unlike females, in which the process of oviposition comes at considerable fitness/energy costs that make them more susceptible to external stressors, male mosquitoes are likely to survive more severe exposure to xenobiotics. It needs to be investigated if males are passing more resistance genes than females to the offspring, through the egg. A straightforward way to investigate the presence of xenobiotic resistance genes in male mosquitoes is to extract their DNA and screen for the presence of known resistance genes.

## CONCLUSIONS

The use of xenobiotics as well as related metabolites, and ultimately their discharge into aquatic environments is a serious threat to those ecosystems. The water solubility of antibiotics and insecticides enhances their uptake through water by various components of the ecosystem. Mosquito eggs depend on water for development, making them very vulnerable to exposure to antibiotics and insecticides. The mosquito egg is very permeable

to antibiotics and insecticides prior to the formation of the SC. The survival of eggs until the formation of SC very much depends on mechanisms that are yet to be investigated. If antibiotics and insecticide resistance genes are indeed vertically transmitted through mosquito eggs, this may aid in the survival of eggs until the formation of SC. This may explain the survival of larvae and pupae until adult mosquitoes emerge. However, the possibility of mosquito eggs harboring insecticide and antibiotic resistance genes raises some serious concerns relative to vector control and public health. Mosquito eggs and larvae are a food source to a variety of fish in aquatic ecosystems, thus the resistance genes in eggs may be transmitted across the aquatic food chain—considering their status near the bottom of the food chain. The mechanisms of resistance to xenobiotics by mosquito eggs, particularly relative to their exposure, needs to be considered due to its potential influence on meeting vector control objectives. Isolation of whole DNA from eggs and screening for known xenobiotic resistance genes is a good starting point. Screening male mosquitoes for the presence of xenobiotic resistance genes helps understand their role in the propagation of resistance.

### Funding
The authors received no funding for this work.

### Competing Interests
The authors declare that they have no competing interests.

### Author Contributions
- Uday Turaga conceived and designed the experiments, performed the experiments, analyzed the data, prepared figures and/or tables, authored or reviewed drafts of the article, and approved the final draft.
- Steven T. Peper conceived and designed the experiments, performed the experiments, analyzed the data, authored or reviewed drafts of the article, and approved the final draft.
- Carlos J. Garcia performed the experiments, analyzed the data, authored or reviewed drafts of the article, and approved the final draft.
- Steven M. Presley performed the experiments, analyzed the data, prepared figures and/or tables, authored or reviewed drafts of the article, and approved the final draft.

### Data Availability
    This is a literature review.

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
