# Peer review of "Xenobiotic resistance in mosquito eggs: current understanding and data gaps"

_PeerJ, doi:10.7717/peerj.19523_

## Round 0.1 · original submission · Major Revisions

The authors should revise according to the reviewers, both of whose comments are constructive, substantive, and valuable.

The paper is well-written, but is marred by minor stylistic flaws. Polishing the paper will make it more readable and will increase its impact. Please consider the following list. For each point, I give one or more examples:

• Omit needless words. Hunt them down and remove them. In line 60, substitute "Mosquitoes are ubiquitous at the" for "Mosquitoes are a ubiquitous presence at the". In lines 133-134, write "the time when the SC forms, at two different", instead of "the time point at which the formation of SC takes place at two different". In lines 233-234, delete "it is important to consider that"
• Don't begin a sentence with a conjunction. In line 47, write "ecosystems, and the" rather than "ecosystems. And the".
• Avoid strings of nouns. In line 188, write "antibiotic-resistant" rather than "antibiotic resistant". In the final sentence of Conclusions, the phrase "One Health concept-driven vector control objectives" strings several nouns together.
• Use specific, concrete terms instead of general, abstract ones. The final sentence of Conclusions is an egregious example of this problem. The point you are making here is important, especially because it wraps up your conclusions. Maybe write two or more sentences to replace this one.
• Scan the paper for typos. I found "Xenoibiotic" on line 184.

Reviewer 1 ·

Basic reporting

The manuscript is generally well-structured and written. However, some areas require refinement for better readability and conciseness. Terminology related to mosquito egg structure and resistance mechanisms (e.g., serosal cuticle, chitin-mediated protection, xenobiotic permeability) could be better defined and explained as to how they mechanistically function in egg biology to improve accessibility for a broader audience. For example, the authors describe genes required for SC formation, but do not describe their functions in cuticle formation and why they are essential. Many aspects of the review provide only a superficial overview of why different aspects of egg development are important mechanistically for resistance to xenobiotics, antibiotics, and insecticides. Providing additional depth as to how different aspects of oocyte biology could confer resistance is needed. Additionally, certain sentences are repetitive, and restructuring some sections would enhance logical flow. For example, The role of SC in protecting against xenobiotics and desiccation is stated multiple times in different sections without significant new insights. A more focused and in-depth section on this would be much more informative and provide the review with a better-defined structure.

One section covering the effects of microbiota on heat stress in adult mosquitoes is never connected to how these findings relate to mechanisms associated with resistance in eggs.

“A microbiome mediated response to heat stress was also observed in adult Ae. aegypti. This involved the enrichment of the midgut microbiome with heat-tolerant taxa such as Bacillus spp. in response to an increase in temperature (Onyango et al. 2020). This suggests a microbiome mediated heat tolerance in adult mosquitoes. The microbiome includes not just communities of microorganisms (microbiota) such as algae, archaea, bacteria, fungi, small protists, but also nucleic acids, proteins, lipids, polysaccharides, nucleic acids, mobile genetic elements, etc. (Berg et al., 2020).”

This section should either be connected to the main topic of the review or removed.

The manuscript incorporates recent and relevant literature to support its discussion. However, the literature search methodology lacks transparency—it would be beneficial for the authors to clarify the number of articles retrieved by their search, any filtering or selection criteria used to curate the retrieved articles, and how studies were prioritized. Additionally, some critical aspects of resistance, such as whether antibiotic resistance genes (ARGs) have been identified in sequenced mosquito genomes, could be better supported with references to genomic databases or studies.
The article follows an appropriate structure with clearly defined sections. However, including a summary graphical figure visualizing mosquito egg structure, potential xenobiotic resistance mechanisms, and knowledge gaps would enhance the manuscript’s clarity and accessibility. The table summarizing serosal cuticle formation timelines is helpful but could be supplemented with additional comparative information across mosquito species, given the diversity of egg-laying strategies and biology between species and genera.
The review is of broad and cross-disciplinary interest, particularly to researchers in vector biology, entomology, environmental toxicology, public health, and One Health disciplines. The manuscript successfully integrates perspectives from mosquito physiology, environmental contamination, and insecticide resistance, making it relevant for multiple scientific communities. However, the One Health framing could be better developed, particularly regarding how mosquito egg resistance may influence disease ecology, food webs, and vector control efforts.
While insecticide and antibiotic resistance in mosquitoes has been previously reviewed, the specific focus on mosquito eggs and xenobiotic exposure is a relatively novel contribution. The manuscript identifies gaps in knowledge, particularly concerning potential resistance mechanisms in eggs and their survival before serosal cuticle formation. Given the increasing presence of xenobiotics in aquatic environments, this focus is justified. However, the manuscript could further differentiate itself by incorporating a more structured comparison of known resistance mechanisms in other insect life stages or species. Additionally, expanding the environmental context of xenobiotic contamination (e.g., sources from agriculture, wastewater, and vector control programs) would improve the motivation for the review.

Experimental design

The methodology for selecting and analyzing the literature is not well-documented, making it difficult to assess the completeness and potential biases in the review.
The manuscript does not specify which databases (e.g., PubMed, Web of Science, Google Scholar) were used or how search terms were combined.
There is no discussion of inclusion/exclusion criteria, raising concerns about potential bias in selecting references (e.g., prioritization of recent vs. older studies, specific study types).
The review mentions various resistance mechanisms but does not clarify whether all relevant perspectives were included, such as genomic evidence from sequenced mosquito species or comparisons with resistance mechanisms in other insects.
The manuscript states that specific topics are underexplored but does not provide quantitative insights into the research volume available (e.g., the number of papers retrieved per search term).
The manuscript needs to: clearly outline search databases and keywords used; define article selection criteria (e.g., time range, study type, relevance); summarize the number of articles reviewed per topic to provide a sense of coverage; compare mosquito egg resistance mechanisms to those in other insect eggs to strengthen the discussion.

Validity of the findings

The manuscript effectively identifies knowledge gaps in xenobiotic resistance in mosquito eggs and presents a compelling case for further research. The discussion aligns well with the goals set in the Introduction, particularly in highlighting the vulnerability of mosquito eggs to environmental stressors before the formation of the serosal cuticle. However, some claims, such as the potential for metabolic detoxification mechanisms or transovarial transmission of resistance genes, lack direct empirical support. While these are important hypotheses, the manuscript would benefit from stronger comparisons to known resistance mechanisms in other insects and a discussion of whether existing mosquito genomic data provides any evidence for these pathways. Additionally, the role of chitin and the serosal cuticle in xenobiotic resistance is discussed in broad terms but could be strengthened with more detail on their specific chemical and structural properties in preventing xenobiotic absorption.

The Conclusion successfully reiterates key knowledge gaps but could be more directly linked to the review’s findings. While the manuscript emphasizes the need for future research, it does not clearly outline how these gaps should be addressed experimentally. For example, transcriptomic or metabolomic studies could help determine whether detoxification enzymes are active in eggs, and genomic analyses could assess whether resistance genes are inherited across generations. Additionally, more discussion on the feasibility of studying these mechanisms in the laboratory versus field settings would enhance the manuscript’s utility for researchers designing follow-up studies. Providing clearer guidance on experimental validation approaches and structuring the Conclusion as a roadmap for future research would greatly improve the impact of this review.

Additional comments

A broader consideration of xenobiotic resistance's ecological and evolutionary implications in mosquito eggs would strengthen the impact. If selection pressures from insecticide and antibiotic exposure influence egg-stage resilience, this could have consequences for mosquito population dynamics and vector control. For example, adaptations in desiccation resistance, cuticular composition, or metabolic detoxification may emerge in response to environmental contamination, yet little research has examined these potential evolutionary shifts. Additionally, if transovarial transmission of resistance genes occurs, mosquito populations may retain resistance traits even without direct insecticide exposure, complicating control efforts. Expanding on these points would enhance the review’s relevance to One Health concerns by considering how human activities may unintentionally shape mosquito resilience and disease transmission potential.

Reviewer 2 ·

Basic reporting

This manuscript aims to review current literature to aid in our understanding of how mosquito eggs are able to survive when exposed to antibiotics and/or insecticides. The review proposes mechanisms for how this may occur and indicates knowledge gaps that should be addressed in future research. Overall, the manuscript is well written and presents a several interesting considerations for researchers to consider in future work.

Experimental design

The article is a review article. The brief description of the methods used to search literature is appropriate in this instance.

Some additional references are required, including sentences beginning Lines 159, 160 and 162.

Validity of the findings

Below are some suggestions that would help balance the manuscript.

1. Title – this implies there will be substantial weight given to the One Health implications of xenobiotic resistance in mosquito eggs. However, there are only a couple of sentences mentioning this. Either remove this from the title and rephrase to: “potential mechanisms and data gaps”. Or, add an additional section to consider the One Health aspect. It would be great to have additional discussion of One Health implications to expand on some of the points briefly touched on in the manuscript. Eg. If mosquito salivary glands contain human gut bacteria with AMR genes, how could these get back to the human gut to transfer AMR?

2. The proposed mechanisms of xenobiotic resistance in eggs, such as vertical transmission of resistance genes in the mosquito genome (without associated bacteria), are presented without supporting evidence. Is there a precedence for this occurring in other insects?

3. It is mentioned several times that eggs do not have a microbiome that could contribute to AMR. Have the authors considered whether bacteria on the exterior of the egg could contribute to reduce the sensitivity of eggs to xenobiotics?

4. Line 152 – I don’t think “attributed” is the correct word here.

5. Section 1 of the manuscript demonstrates the vulnerability of eggs to xenobiotics prior to SC formation. However, it does not provide evidence that susceptibility of eggs to xenobiotics are changing over time. Therefore, Line 180 – this should state it’s not clear IF eggs are evolving to combat exposure to xenobiotics.

6. Line 200-203 – please reorder these sentences and start with the viability sentence first.

7. Lines 205-211- should add the caveat that ARGs from proteobacteria would need to get from skin to gut for this to be reciprocal transfer

8. Line 252 – this language should be softened – “one possible explanation”

Additional comments

None

---

## Round 0.2 · Major Revisions

While your revisions improve the paper, you did not make all the major revisions requested by the reviewers. You should make a list of all the revisions requested by each reviewer, and then either make the requested changes or give good reasons why that is not possible or why you disagree with the reviewer on that point. I'll give two examples: Reviewer #1 asked you to define and explain terminology related to mosquito egg structure, and to explain mechanistically how they function. Also, he requested that you prepare a figure showing egg structures and listing the various resistance mechanisms as part of that diagram/figure. In this case, even a table would be enhance the presentation. I await your next revision.

---

## Round 0.3 · accepted · Accept

The inclusion of a "rebuttal" was helpful.
Now your review is clear and compact.